# Monkeypox Virus: A Comprehensive Overview of Viral Pathology, Immune Response, and Antiviral Strategies

**DOI:** 10.3390/vaccines11081345

**Published:** 2023-08-09

**Authors:** Shiza Malik, Amna Ahmed, Omar Ahsan, Khalid Muhammad, Yasir Waheed

**Affiliations:** 1Bridging Health Foundation, Rawalpindi 46000, Pakistan; shizzza874@gmail.com; 2Department of Oncology, Jinnah Hospital, Lahore 54550, Pakistan; c26.amnaahmed@gmail.com; 3Department of Medicine, Foundation University Medical College, Foundation University Islamabad, Islamabad 44000, Pakistan; omar.ahsan@fui.edu.pk; 4Department of Biology, College of Sciences, UAE University, Al Ain 15551, United Arab Emirates; 5Office of Research, Innovation, and Commercialization (ORIC), Shaheed Zulfiqar Ali Bhutto Medical University (SZABMU), Islamabad 44000, Pakistan; 6Gilbert and Rose-Marie Chagoury School of Medicine, Lebanese American University, Byblos 1401, Lebanon

**Keywords:** monkeypox virus (mpox), outbreak, poxviruses, vaccines, therapeutics, immune responses

## Abstract

Background: The years 2022–2023 witnessed a monkeypox virus (mpox) outbreak in some countries worldwide, where it exists in an endemic form. However, the number of infectious cases is continuously on the rise, and there has been an unexpected, drastic increase in cases that result from sustained transmission in non-endemic regions of the world. Under this scenario, it is pertinent for the world to be aware of healthcare threats to mpox infection. This review aimed to compile advanced data regarding the different aspects of mpox disease. Methods: A comprehensive strategy for the compilation of recent data was adopted to add data regarding mpox, biology, viral pathology, immune response, and brief details on the antiviral strategies under trial; the search was limited to 2016–2023. The aim is to make the scientific community aware of diverse aspects of mpox. Results: Consequently, detailed insights have been drawn with regard to the nature, epidemiology, etiology, and biological nature of mpox. Additionally, its host interaction and viral infectious cycle and immune interventions have been briefly elaborated. This comprehensively drawn literature review delivers brief insights into the biological nature, immune responses, and clinical developments in the form of therapeutics against mpox. This study will help scientists understand the biological nature and responses in hosts, which will further help clinicians with therapeutic handling, diagnosis, and treatment options. Conclusions: This study will provide updated information on mpox’s pathology, immune responses, and antiviral strategies. Moreover, it will also help the public to become educated on the healthcare-associated threat and take timely mitigation measures against expected mpox outbreaks in the future.

## 1. Introduction

Countries around the world have been undergoing an immense struggle against the healthcare burden put forth by the COVID-19 pandemic [1]. People have slowly adapted to its permanent prevalence in the form of a “new normal”, as the virus was reported to evade eradication on a global scale, even if vaccination is ensured worldwide [2]. During this healthcare burden, another viral disease has arisen in the form of infections caused by mpox [3]. Although the virus has been known for almost half a century, its infections were previously limited to African countries [4]. However, in the recent past, the frequency of mpox outbreaks and infectious cases has been increasing for an extended period, both in endemic and non-endemic regions.

Mpox was discovered and identified in 1958 in the form of an outbreak within an animal facility in Denmark [5]. In medical history, the first mpox infection was reported in the Democratic Republic of the Congo in 1970. Meanwhile, several other species of mpox have been discovered and found mainly in African countries [3]. The recent upsurge in 2022 has spurned the scientific community to look into the causes and concerns of mpox outbreaks regarding the disease’s origin, natural history, biology, epidemiology, healthcare implications, and ways to deal with it [6]. Moreover, since the WHO has declared the mpox outbreak a global health emergency, this puts pressure on the scientific community to expand understanding and research work on the mpox threat [7,8].

Although much has been uncovered with regard to the aspects of mpox biology, structure, host–immune interaction, evolutionary characteristics, epidemiology, transmission characteristics, and ecological dimensions, a deep sense of understanding remains to be developed for this class of viruses [9]. It is important to develop an understanding of their infection in humans and to discover methods for clinical handling of the disease, which may include therapeutic and vaccination designed specifically for mpox-related disorders [10]. Therefore, monkeypox-related studies have been conducted, mostly in 2022 [9]. The purpose of this compilation is to bring to the attention of the scientific community a brief overview of mpox-related virology, immunological responses from hosts, and therapeutics that are given consideration to fight mpox infection. Moreover, this study will highlight the gaps in research that require further understanding so that in the future, more advanced biotechnological conclusions can be drawn related to mpox virology and better therapeutics can be proposed for handling virus loads in healthcare systems [10].

## 2. Methodology

### 2.1. Search Criteria

The data for this comprehensive review have been gathered from the most diverse research journals, recent publications, and most cited clinical studies related to monkeypox virus. The search strategy acquired for this purpose has been a systematic one to gather data from electronic sources, including mainly Google Scholar, while some other platforms like PubMed, NIH (National Library of Medicine), and Web of Science, European database, Springer, and Embase databases have also been given consideration. Moreover, the official websites of WHO and UNAIDS have also been used to obtain the statistical results and latest updates regarding the MPXV outbreak and epidemiological episodes in recent years. As this study mainly covers mpox-related viral pathology, immunogenicity, and antiviral strategies, the major research items were viral pathology, immune response, and antiviral strategies; outbreak; poxviruses; vaccines; therapeutics; immune responses, and some other linked search terms.

### 2.2. Inclusion and Exclusion Strategy

The inclusion and exclusion criteria simply revolved around gathering data from original research articles, sections from books, letters to the editors, short and lengthy reviews, and some case studies published recently. The available data were immense, but we tried limiting the data to around 50 studies, from which the interlinked data have been gathered. Thus, after a thorough analysis of the dates, abstracts, titles, and journals of research publications, they were included in this review. Moreover, this review has mostly reviewed studies conducted in last five years (2018–2023) to add only the most recent advances related to MPXV updates, especially during the latest endemic episodes of 2022 and 2023. It should be noted that studies of English origin have been made part of the review.

### 2.3. Structuring the Article

The review was structured to initially incorporate the biology and natural history associated with mpox and related viruses, along with a brief discussion about the etiology, structure, replication, and epidemiological records of mpox. In the later sections, the clinical manifestations, pathogenicity, latest clinical case reports, and human studies are elaborated to present how the virus presents danger to human hosts. We also highlight the mpox host interactions, viral transmission, and life cycle, along with predictions of mpox host immune system interaction with monkeypox. In the last portion, we briefly discuss some potential vaccines, therapeutics, and antiviral strategies being devised against mpox, along with some related zoonotic poxviruses that affect humans. In the discussion portion, we focus more on the current mpox outbreak (2022) and future recommendations to deal with the expected outbreak of mpox in both non-endemic and endemic regions.

## 3. Biology and Natural History

Mpox is a zoonotic virus that is transmitted between humans and animals. This virus belongs to the genus *Orthopoxvirus* (this genus also includes variola (smallpox) and Vaccinia viruses), all of which belong to the family of Poxviridae. It is a diverse and widely distributed family of viruses. The separate types of mpox virus are referred to as clades or subclades. The recent 2022 outbreak has been accounted for by clade II of mpox, which is historically named the West African clade, or more specifically, the subclade IIb. Mpox multiply within the cytoplasm of their host cells with the help of their double-stranded (ds)DNA. The role of the host cell nucleus, however, remains elusive within the replication process [11]. Two subfamilies of Poxviridiae have been defined according to their host interactions: *Entomopoxvirinae* and *Chordopoxvirinae.* The latter subgroup *Chordopoxvirinae* mostly infects vertebrates and is further subdivided into 18 genera, whereas the genus Entomopoxviriane infects invertebrates and has a further 4 genera.

Because mpox belongs to the family of *Poxviridae*, they exhibit the same biological characteristics as other *Orthopoxviruses*. They are ovoid-shaped particles ranging in size from 200 nm to 250 nm. The outer envelope is composed of a lipoprotein membrane encompassed by a capsid. Some distinct surface tubules with a well-established dumbbell-shaped core are part of its structural composition [4]. Just like other virus types, the envelope functions to protect capsids and nucleic acid regions that contain all the necessary chemical components, such as enzymes, dsDNA of around 200 kb, transcription factors, virus–host interaction genes, and housekeeping genes. On a broader structural level, the central and terminal regions are defined by the encoding of housekeeping genes in a conserved region and the presence of host interaction proteins in a variable region. Despite the presence of these structural components, mpox still relies on host ribosomal bodies for translating viral mRNA [12].

## 4. A Brief Overview of Etiology, Structure, and Replication

Unlike the name, monkeys are not the natural hosts of mpox; instead, rodents, rats, and squirrels, among other related animals, are considered their principal reservoirs. That said, infections have also been reported in non-human primates such as monkeys (WHO, 2022a) [13]. Worldwide large-scale investigations are ongoing regarding animal-to-human transmission of the virus. Mpox was initially found to be an endemic disease limited to Africa [3]. Although it was already known to the scientific community (identified in the mid-20th century), mpox infection slowly gained identification in the form of a zoonotic endemic disease among the general community when reported in widespread regions worldwide [13]. The exact case prediction for mpox infections is not confirmed because scientists believe that its diagnosis has created some confusion due to its similarity with smallpox infections. However, after the 1970s, confirmed diagnosis of mpox cases was reported, and afterward, African countries continuously witnessed episodes of mpox infections [4]. The major cause for the outbreak of most zoonotic diseases such as Ebola, Dengue, COVID-19, and mpox infection from the African region may be due to the dependence of a large-scale poor community on the hunting and eating of animals along with the international travel and import–export of infected animals to a non-endemic region, although the ongoing global outbreak of subclade IIb has no specified zoonotic link associated with it and requires further investigation regarding the infection’s spread [13]. This aspect has raised alarms in the scientific community to coordinate a comprehensive strategy to tackle this healthcare load without compromising the region economically [13].

## 5. Epidemiological Records of Mpox

Statistical and epidemiological data from different studies help trace records of mpox outbreaks and transmission in past history, such as clade I emergence in DRC, and mpox clade IIa emergence in the US back in 2003, which might have been transmitted from Ghana, as indicated by epidemiological investigations [3]. Similarly, other mpox outbreaks with a limited number of infections were reported in areas of Sudan and Nigeria back in 2017 and 2018 [3]. Some cases have also been reported in non-endemic countries such as the United Kingdom, Israel, and Singapore. According to recent reports by the WHO, thousands of cases were reported in only one country: the DRC in 2022, in which approximately 4600 new cases of mpox, with 36 deaths, were recorded in the month of October 2022. Additionally, the outbreak spread to 109 countries worldwide until the aforementioned month [14]. The European Centre for Disease Prevention and Control (ECDC) reported some 20,675 confirmed cases identified in 29 EU nations by the end of October 2022 [15]. Moreover, hundreds of other infection cases have been reported in Liberia, Sierra Leone, Cameroon, Congo, the Central African Republic of Nigeria, and the DRC. These incidences indicate the global immigration of mpox from endemic to non-endemic countries, indicating a threat to healthcare [16].

As of the start of 2022, WHO reported mpox cases from 112 member countries. The last update provided by WHO, cited on 10 July 2023 reported a total of 88,288 confirmed cases of mpox infection and 1084 suspected cases, as well as a death toll around 150 [17]. These cases have not been limited to the pandemic countries like West and Central Africa, as they have been reported from a number of countries without a previous record of mpox transmission. In most of the cases, the reported persons are men who have sex with men (MSM) [17,18]. However, the nature of the sexual transmission of mpox disease is yet to be verified. The confirmation of mpox cases in different countries is considered an outbreak in those regions, and therefore, proper recommendatory measures should be taken swiftly to avoid a pandemic-like situation like that in African countries [17]. Moreover, it should be kept in mind that the WHO has assessed the global risk of mpox as low to moderate in most of the regions of the world as the frequency of reported cases is decreasing substantially over time. Therefore, the mpox outbreak is no longer considered a public health emergency of international concern [19]. Thus, by following the temporary recommendations for the transitionary period, the long-term control of mpox could be adopted to overcome the expected healthcare burden linked with mpox [17,20].

## 6. Clinical Manifestations and Pathogenicity

The clinical features of mpox infection appear mainly in the form of skin rashes and fever. The fever remains asymptomatic for an incubation period of 10–21 days, in which it increases up to 38–40 °C. During this period, cutaneous dermal petechia occasionally develops. Some other clinical manifestations may involve headaches, body aches, nausea, vomiting, prostration, fatigue, and weariness that develop after at least 1–3 weeks [21]. A distinguishing feature of mpox infection is the development of swelling at the maxillary, cervical, or inguinal lymph nodes (lymphadenopathy) that, in later stages, causes extreme difficulty in drinking and eating [22]. Moreover, a widespread systematic rash develops 1–2 days after infection. The rash feature is particularly restricted to the oral mucosal surfaces and the face from where the fever abates. The rashes on cutaneous lesions are similar to smallpox lesions [12]. These lesions progressively extend from macules, papules, and vesicles to pustules in any region of the body and toward crusting and scarring. During rash remission from these regions, hypopigmentation accompanied by hyperpigmentation appears in scarred lesions. Some clinical features that differentiate symptoms of smallpox infection from mpox infection may include severe cervical, postauricular, submandibular, and inguinal lymphadenopathy in mpox infection [23].

Higher lesion counts lead to severe complications of respiratory and gastrointestinal tracts, such as encephalitis, septicemia, and ocular infections, which may even lead to permanent vision loss. The virus infection is more severally manifested in the upper middle and lower air tracts due to them being the prime routes of infection [24]. Skin rashes may lead to dermal bacterial infection, particularly in unvaccinated patients with small poxvirus vaccines. In pregnant women, mpox infection vertically transfers between the mother and fetus. The chances of healthy childbirth are minimal. Similarly, studies on children indicated a fatal version of mpox disease with a higher fatality rate compared with older individuals. This disparity may be directly related to the weaker immune system in children [25].

The process by which infection from tissue-resident immune cells (monocytes, macrophages, B-cells, and dendritic cells) progresses in the body remains a matter of in-depth debate for future vaccine implications. However, it is a matter of common observation in mpox infection that the lymphatic system acts as the primary source of viral dissemination, as it replicates extensively in the lymphoid tissues of the head and neck regions [26]. Mouse model-based studies revealed that the lymphatic system, spleen, and liver have been the major targets of viral infection, from where viremia slowly spreads to the lungs, kidneys, intestines, skin, and other vital body organs. Although studies support viral infection and replication in the lymph nodes, spleen, and bone marrow, the question of how the infection progresses from these routes remains elusive and needs further investigation [27].

## 7. Clinical Case Reports and Human Studies

Although several studies have been conducted on animal models, there is a need to gather data from cases of real patients for studies of mpox-related implications. Apart from studying the clinical manifestations, scientists have gathered data regarding the implacability of the smallpox vaccine on mpox patients, especially during the recent outbreak [28]. The concept of cross-protective immunity has been studied extensively during these cases. Surveillance data from the Democratic Republic of Congo (DRC) community indicated that mpox attachment rates are higher in individuals who lack a history of smallpox vaccination. The same vaccine provides 85% efficiency for patients in some places [29].

In one study conducted in Peru, the highest number of mpox cases per million was reported in Latin Americans, with concentrations found in men having sex with men. Moreover, a viral co-infection has been reported in these patients, with approximately two-thirds of Peruvian patients suffering from mpox and HIV. In these patients, clinical symptoms of skin lesions in the anogenital area without preceding systemic symptoms were found to be common. Moreover, the majority of hospitalized mpox cases had HIV and were on antiretroviral treatment [30]. In most of the previous studies, patients reportedly had symptoms, including rashes and anogenital and mucosal lesions, alongside systemic features like fever, lethargy, myalgia, lymphadenopathy, and headache. In most of these cases, deaths were not reported unless otherwise suffering from concomitant viral infections [31]. Occurrences in bisexual individuals have also been reported in some other studies. However, it is still not established whether concurrent sexually transmitted infections (STIs) contribute to mpox infection or control its clinical expression. However, concomitant infections such as syphilis and mpox and HIV and mpox show that there might be a correlation among the pathogenic infections. This inference needs further scientific support for confirmation [32,33].

In some recent studies conducted by scientists, the clinical data from cases of monkeypox infection were derived from a single center in Italy [34,35]. The majority of cases were imported from foreign countries, with the most common symptoms being fever, rash, and lymphadenopathy. Respiratory symptoms were also present in a small number of cases [34,35]. The study found that monkeypox can present with a wide range of clinical features and that diagnosis can be difficult due to its similarity to other viral illnesses [36]. Moreover, in the context of the current outbreak, atypical cutaneous features have emerged (maculopapular rashes; a controversial relationship with syphilis) and new supportive diagnostic modalities have been studied (namely, dermoscopy) [35].

These studies suggest that initially, the treatment is primarily supportive, although antiviral drugs may be beneficial in severe cases. Some studies also predicted the incidence of mpox in patients with previous vaccination with smallpox. These, along with some other studies, helped scientists infer that a history of smallpox vaccination may provide protection against mpox infection but not necessarily protect against symptomatic disease [11,37,38]. Patients with previous smallpox vaccination develop a less extensive rash, fever, and lesions compared with unvaccinated individuals. However, with genomic mutations and changes in transmission modes, the need to create separate vaccination protocols for mpox has been of importance to the scientific community [38].

## 8. Host Interactions, Viral Transmission, and Life Cycle

Studies on the biology and host interaction of mpox are helpful for molecular virologists to find other linked disease-causing viral species and discover new approaches for therapeutics and drug targets for mpox and the related family of viruses. Conformational studies are still ongoing to pinpoint the exact routes of viral transmission in humans. A brief insight into the viral infectious cycle elaborates that close physical proximity with infected people may play a vital role in transmission [33]. Mpox may enter through multiple routes into the host cells such as face-to-face exposure, bodily fluids, shedding through feces, entry through the (nose) nasopharynx, (oral) or pharynx, (muscular) intramuscular, (skin) subcutaneous, mucosal surfaces, and/or intradermal routes. Micropinocytosis, endocytosis, and fusion processes are used for entry into host cells with subsequent replication at the site of infection [4]. This is followed by the activation of immune responses from the host, either in the form of phagocytosis or necrosis. From the bloodstream, the virus takes entry into the lymphatic system and starts to affect other vital organs, such as lymph nodes, spleen and tonsils, and bone marrow. Slowly, it multiplies and exhibits clinical manifestation in the form of mpox-related disease in the host [39].

Additionally, studies have revealed that infections of mpox are differentiated into two versions due to the cause of two distinct forms of the virus: mature virion (MV) and enveloped virion (EV), which differ depending upon membrane layers and surface glycoproteins [4]. MVs enter via micropinocytosis or fusion processes into host cells, whereas EVs undergo the removal of the outer membrane before fusing with host cells. Both forms then undergo a replication process within the host cytosol to release viral genomes and respective viral proteins within the cellular atmosphere. The expression of early genes is followed by intermediate genes and transcriptional regulatory factors that result in viral dsDNA [40].

The replication, transcription, and translation mechanisms help assemble intracellular mature virion (IMV) after passing through the stages of immature viral DNA formation. IMV then acquires outer wrapping from the Golgi apparatus to adapt into intercellular enveloped virions (IEVs) [12]. IEVs may undergo further fusion with the inner cell membranes to form cell-associated virions (CEVs), which, upon release from the cell, are called extracellular enveloped virions (EEV). These viral forms, such as IMV, IEV, and CEV, are ultimately released from the host cell to interact and infect neighboring cells via host cell lyses. During this release, they develop an extra membrane before exiting the host cell. These events are fabricated to explain the intricate replication cycle of mpox [12,40].

## 9. Mpox Host Immune System Interaction Prediction Based on Poxvirus Infection Similarity

Immune responses from host cells in animal and human models have not yet been established, even after decades of viral exposure to humans. Perhaps the virus remained less contagious and had limited infection rates up to this outbreak. After the declaration of the global health emergency (2022) of mpox, the scientific community largely became involved in figuring out the routes to deal with this healthcare emergency. Therefore, some insights are being drawn into the immune–virus responses against Vaccinia virus (VACV) and related viruses of the family Orthopoxviruses to improve our understanding of mpox. These viruses share the same process of infection, which is confirmed by similar genes and encoded proteins that interfere with host cell signaling responses, which ultimately reflect disrupted viral recognition, apoptosis, and immune responses. Mammalian cells can easily detect microbial interventions through pattern recognition receptors (PRRs) that undergo cellular signaling to activate transcription factors such as NF-κB and interferon regulatory factors (IRFs) and cellular and humoral responses from innate and adaptive immunity, which play a vital role in fighting foreign entities.

### 9.1. Innate Immune Responses

The first line of defense in the form of innate immunity follows an active viral infection in the body, although there are instances where the same route provides a viral entry route. Monocytes have been identified as the major route of poxvirus entry as antigen representation by monocytes, and neutrophils predict the lethality generated by mpox [37]. Studies on animal models revealed that these monocytes are recruited to the sites of infection and lead to a marked expansion of CD14+ monocytes. Similarly, in mouse models, inflammatory monocytes have been predicted to be a possible vehicle for viral distribution. Similarly, human-based experiments showed that primary M2-like macrophages allow VACV replication and distribution via processes such as the formation of cell linkages, branching structures, and actin tail formation associated with virus spread [41]. This does not limit the spread mechanisms of VACV as deletion of these phagocytes does not eliminate viruses from the host cell, indicating that certain other immune cells might also be involved in viral dissemination. Moreover, both neutrophils and monocytes are involved in limiting viral-infection-associated tissue damage. All these studies point to the linkage of adaptive responses or a second line of defense to control viral spread in the host body [42]. Although a great level of similarity exists between VACV and mpox, studies are still needed to confirm immune system responses toward mpox infection in different subject models.

### 9.2. Adaptive Immune Responses—B-Cell-Mediated Antibody Protection

The importance of B-cell-mediated immune responses in the form of the production of specific antibodies is of prime importance due to the successful vaccination of smallpox in history based upon B-cell-mediated immunity [43]. Some studies have suggested that treatment with vaccine immune globulin (VIG) could save patients from mpox infection [44,45]. This hypothesis was checked in monkey models and showed that VACV-specific B-cell responses could protect against a lethal mpox infection [43]. This sort of inference is beneficial for the scientific community designing vaccines along similar lines to the VACV vaccine or using VACV against mpox infection to check the results [44]. Studies conducted on this cause revealed increased production of memory B cells and neutralizing antibody loads. Approximately 14 proteins of mpox are recognized by immunoglobulin produced by B-cell responses from human vaccination protocols [46]. These proteins (namely, D8, H3, A26, A27, C19, A33, A44, c7D11, c8A, L1, and B5, among important names) have functional characteristics and involvement in the infection and immune-regulatory responses associated with mpox [29]. These proteins hold the potential to be used as serological diagnostic markers. If we talk about the roles of specific antibodies, IgM plays a vital role in the primary immune response, which is comparatively less effective than the IgG-mediated secondary immune response against mpox [47]. These studies specifically highlight the need to extensively study antibody profiles in patients with mpox.

### 9.3. T-Cell Immunity Responses

As elaborated earlier, CD4+ T cells play a vital role in immunity in terms of mediating functional memory B-cell responses. These cells are related to the production of IFN-γ and TNF following vaccination with VACV. A study of patients with HIV has shown that a compromised CD4+ cell count leads to higher risks of developing a severe mpox infection. The results were more explanatory when studies on patients with HIV with higher CD4+ cell counts were observed to lack severe mpox infection development [6]. Nevertheless, more insights into the observations are needed. Additionally, T cells provide direct antiviral responses besides activating antibody release. CTLs have been found to directly kill infected macrophages and prevent further viral spread. Similarly, CD8+ T cells eradicate virus-infected monocytes to control viral loading along with the activation of IFN-γ, which proves to be effective against viral lethality even in the absence of B cell and CD4+ T-cell responses. Similarly, cytokines such as IL-1β, IL-1RA, IL-2R, IL-4, IL-5, IL-6, IL-8, IL-13, IL-15, and IL-17 expression increases following mpox infection in humans. This cytokine profiling indicates dominant helper T-cell and regulatory T-cell responses in an inflammatory environment [37]. In summary, T-cell responses in controlling *Orthopox* viral infection are significant and could be considered for vaccine design. However, comprehensive and confirmatory studies are necessary to correlate the relationship between CD4+ and CD8+ T -cell responses with the severity of mpox infection in humans.

## 10. Monkeypox and Smallpox (Vaccinia) Viruses Correlation and Vaccine Development

There is a certain level of similarity between viral biology; infection cycle; transmission mode; and clinical manifestations of VACV, mpox, and other members of the family Orthopoxviridae [2,29]. Specifically, the antigenic similarity between smallpox (Vaccinia virus) and mpox could be used to manufacture vaccines and antiviral pigments against the mpox epidemic [2]. Some details regarding the similarity of host and immune system interactions and responses for VACV and mpox have already been explained. The clinical effectiveness of approximately 85% has been recorded for the smallpox vaccine against mpox infections [2]. These data have been largely supported by animal-based studies. This pertains to the use of the smallpox vaccine against mpox in humans. However, the smallpox vaccine comes with some side effects, and thus more scientific experiments are formulating drug replications that are effective for mpox and have reduced side effects [6]. These interpretations have yet to be completely determined in patients with mpox for future implications. Since the immune system can recognize and respond to multiple similar Orthopox viruses, similar immune responses could be helpful in preventing infection from these viral species [48]. The characteristic cross-reactivity of the immune system is a feature used by scientists to check animal models with smallpox infections to analyze vaccine and drug formulation efficacies against other-linked viral diseases as well as those caused by mpox [44].

## 11. Potential Vaccines, Therapeutics, and Antiviral Adjuvants—An Update

Some clinical studies show that vaccines for the smallpox virus family also have functionality against mpox [29]. Some vaccines are approved by the FDA for pre-disease vaccination against mpox. One of these vaccines is the live VACV vaccine ACAM2000, common in the market and an approved smallpox vaccine that has shown some immune responses in mpox-infected patients. Following VACV vaccination, patients have developed immune responses in the form of activated CD8+ T-cells and higher levels of the degranulation marker CD107, along with TNF, IL-2, and CCL4, which all together lead to a wide spectrum of T-cell responses [49].

Another therapeutic intervention has come forward in the form of MVA, a Vaccinia virus strain Ankara-based vaccine also named JYNNEOS, which is also composed of live attenuated virus particles and belongs to a third-generation vaccine and shown elevated CD8+ T-cell responses compared to first-generation Drax immunization [50]. Both ACAM2000 and Dryvax have been experimented with, and it was found that they produce VACV-neutralizing antibodies upon vaccination. It is worth mentioning here that the Dryvax vaccine has been extensively used during the smallpox eradication campaign and is therefore being considered for mpox eradication efforts. JYNNEOS has also been shown to have a better safety profile compared to other vaccines and develop almost similar neutralizing antibody titers as those of the first-generation ACAM2000 [51]. However, its dose usage need and interval are still undetermined [51,52].

Although these vaccines are advantageous in providing a certain grade of immunity, most of these vaccines are associated with some side effects, such as myocarditis and pericarditis, especially in vulnerable groups such as those with eczema and pregnant ladies [25]. Factors such as days post-vaccination; particular subject, i.e., animal model or humans; time and dosage; rate of exposure; immunocompetency; and type of vaccine also matter. The vaccine design is critical to be well understood and well defined with the appropriate knowledge of the virus’ pathogenic and mutational potential. The formulation of a single-shot path as an effective therapeutic is the aim of scientists to ensure public health and cost benefits against virus pandemics [53]. Knowing this, scientists have designed some other vaccines besides VACV or MVA-based vaccines for poxvirus strains, such as NYVAC and ALVAC, but they must establish further studies to understand the efficacy and effectiveness of these newer vaccine candidates against mpox infection. Moreover, further clinical trials are needed to register and license vaccines in the market.

## 12. Therapeutic Options

Similar to the matching vaccination strategy between smallpox and mpox, some therapeutics developed for smallpox are also used against mpox infections. One such antiviral drug is tecovirimat, which does not affect the mature virus, but affects its lifecycle by targeting the VP37 membrane protein of mpox, inhibiting the maturation of the enveloped virus [38]. Their effectiveness has been determined against NHP models and remains to be well-evaluated in humans.

Another important antiviral drug is brincidofovir and its active ingredient, cidofovir, approved by the FDA, which works by inhibiting *Orthopox* virus DNA synthesis to prevent viral replication. It is approved for smallpox vaccination but is efficient against a large number of dsDNA viruses due to a similar notion of replication inhibition. Its efficacy has been determined in dog and mouse models, but its cytotoxicity and human experimentation remain to be determined [54]. Similarly, another drug, VIG, has been used against smallpox and has shown cross-neutralizing activity against mpox in monkeys. VIG needs to be evaluated for mpox in both animal and human models [55]. These vaccination protocols help provide better management options for mpox infection. For years, management was limited and relied upon supportive care and handling of health complications; however, with the evaluated efficacies of the smallpox vaccine against mpox, the generation of new antivirals has opened a door for scientists to discover new therapeutics [56].

## 13. Other Poxviruses That Affect Humans

The sudden outbreak of mpox highlights the need to look for other orthopox viruses that could cause severe human infection and could be a threat to the healthcare system. It is also necessary to be proactive and design drugs and vaccines before these diseases infect humans at a mass scale. Moreover, the understanding of viral, biology, ecology, epidemiology, etiology, and immune and host reaction will be helpful in the design of further antiviral drugs and to determine their co-efficacious potential [57]. Viruses such as ectomelia (mousepox) and camelpox (mainly associated with infection in camels) have already placed a socio-economic burden, and therefore, it is important to determine the pathological nature of other relevant viruses to avoid future infection incidences and to lessen the burden on healthcare and the economy as they incur considerable loss in terms of morbidity and mortality in camels, loss of weight, and reduction in overall milk yield, especially in camel-rearing countries [58]. Some other known zoonotic species of poxviruses that could cause infection in humans and could be a healthcare burden include Cowpox/Buffalopox (*CPXV*), Variola (smallpox), Vaccinia virus (*VACV*), Molluscum contagiosum virus, and Orf virus [58].

Vaccinia virus (VACV) is the most prominent in this regard and causes severe infection in humans if the person is unvaccinated. Mpox, as explained in detail, is quite similar to variola in terms of structure, functional characteristics, and disease manifestation [29]. Similarly, the genus paradox, orf virus, pseudocowpox virus, bovine papular stomatitis virus, red deerpox virus, and grey sealpox viruses cause a certain level of infection in humans. Although their prime hosts are animals, they can cause infections in humans. Furthermore, the genus Yatapoxvirus has two viral species with zoonotic characteristics, namely, Tanapox virus and Yaba monkey tumor virus, which may lead to zoonosis in humans with the potential to spread by natural processing. In the case of the genus Molluscipoxvirus, a single species named molluscum contagiosum virus (MCV) causes zoonotic infection, but their human disease is yet to be determined in detail. MCVs are similar to poxviruses due to the similarity in replication mechanisms [59].

## 14. Current Mpox Outbreak (2022) and Future Recommendations

The outbreak of mpox (2022) is here, and the strategies for dealing with the viral spread should be well known in the scientific community and the public. This may include clinical trials and laboratory assessments, contact tracing, and effective social control measures. Healthcare authorities such as the WHO and CDC have presented some recommendations for dealing with mpox disease containment, which may include precautions regarding diagnosis, treatment, and patient isolation [59]. Moreover, some recommendations have been put forth for the use of the approved vaccines ACAM2000 and JYNNEOSTM in the form of pre-exposure among high-risk groups and booster doses for mpox patients [51]. Similarly, some protocols have been outlined, which state that researchers and lab technicians working in a laboratory need to have proper laboratory instruments, sample handling, and occupational regulations [60].

Moreover, there is a continuous need for researchers, public health workers, government authorities, and the public to coordinate in the form of a participatory approach to handle the further spread and work for mitigation and adaptation to the mpox epidemic. Affected nations must establish strategies for endemic handling to ensure global safety. For this reason, there is a need for international collaboration for training, prevention diagnostics, surveillance, and treatment strategies to avoid endemic conversion to a pandemic like that of COVID-19, which has been quite recently experienced.

## 15. Discussion

Recent outbreaks of infectious diseases such as mpox, dengue, ebola, and others have forced scientists to work on therapeutic strategies in more profound ways than ever before. For the scientific community, the past three years have been particularly crucial in terms of treatment strategies and vaccination protocols against such viruses, especially during the pandemic era. This is the very reason that so much research is being conducted on vaccination and antiviral drug protocols against such viral diseases. This literature contains several vaccination protocols and therapeutic strategies for mpox, which indicate that such trials can lead to improved diagnosis of mpox disease, improved treatment options, and an overall reduction in healthcare costs and associated economic burdens in mpox-affected regions. Mpox may present some hurdles for clinical trials due to its infectious nature, lack of funding, and poor coordination among clinicians and scientists in endemic countries.

Under the described scenario, it is pertinent to encourage clinical experiments up to levels III and IV to obtain licensure vaccination in the market. Limited lab resources and technical glitches caused by a lack of clinical data are causing experiments to be delayed. Developed nations are well acquainted with the importance and maintenance of healthcare measures. There is a need for these advanced nations to keep on working with vaccine development in their special advanced labs and conduct studies on human subjects to establish statistical data in support of vaccine candidates. Furthermore, they should consider modern therapeutic approaches, such as those based on nanotechnology, to develop new vaccines and drugs. In developing nations, special focus should be kept on adaptive strategies to cope with mpox. Further confirmation of existing vaccination protocols should be sought, as well as incorporating some of the latest technologies into existing vaccination drives to create improved vaccination protocols, such as plant-driven organic compounds and nanomedical applications.

In the future, standard methods of vaccination such as vector-based vaccination, live attenuated protocols, subunit vaccines, and others are likely to be advanced in terms of manufacturing techniques such as a combined application of gene-splicing, genetic engineering protocols, and nanobiotechnology, which can be helpful to enhance therapeutic manufacturing, dosages, outcomes, and clinical experimentation compared to previous drug protocols against mpox. In the next five to ten years, there is a need to provide more research funding to implement advanced research protocols and wide-scale clinical experiments against mpox. Moreover, there is a need to indulge the general population, healthcare workers, and educated populace in participatory approaches to health management and social awareness. This will allow endemic countries to adopt better mitigation and adaptive practices against mpox and related healthcare practices. Nonetheless, more scientific experimentation should continue to tackle the surge of mpox and similar infectious diseases to lessen the burden of the global healthcare system.

## 16. Conclusions

The current outbreak of mpox is a reflection of the longstanding pathogenicity of poxviruses against humans. Rapid spread, alongside cross-genomic mutations, is a major threat to this endemic that may increase its intensity. It is critical to develop insights into mpox biology, pathological dimensions, epidemiology, rates of infection, and therapeutic hurdles to develop a robust strategy for handling the outbreak. Moreover, as the disease is still in its endemic state, it is better to devise and practice a broad-scale immunization drive in endemic regions to tackle the outbreak in its current form and prevent its further spread. Further, there is a threat of increased transmissibility, genomic changes, and augmented virulence exits. To deal with these risks, it is pertinent to ensure the availability of currently approved vaccines exclusively in endemic regions. In addition, there is a dire need to strengthen the healthcare management capabilities of affected countries to deal with this emerging epidemic. Moreover, the scientific rigor to discover preventive medicines and vaccines should be continued to discover the most effective therapeutic interventions to deal with an mpox outbreak. It is important to couple healthcare practices with public cooperation to create awareness, mitigation, and adaptation patterns against present and future outbreaks of a similar nature.

## Data Availability

All the data is present in the manuscript. Additional details can be provided by writing email to corresponding author.

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
