# Peer review of "Monkeypox Virus: A Comprehensive Overview of Viral Pathology, Immune Response, and Antiviral Strategies"

_vaccines, 2023, doi:10.3390/vaccines11081345_

Round 1

Reviewer 1 Report

The comment added as sticky notes in the attached pdf

Author Response

The comment added as sticky notes in the attached pdf

Response: all the asked changes were mostly related to formatting and have been made in the document.

Reviewer 2 Report

The review presented here is all about providing the latest updates regarding the MPX infection.

It appears somewhat superficial and the bibliography is not well structured.

In fact, in several cases the references cited in the text do not correspond to the studies listed among the references. In other cases, references are completely missing (for istance , see lines 145-149).

The section regarding therapeutic options is too concise.

The authors should add more data about the last epidemiological scenario of MPX.

Finally, it would be more appropriate to describe virology and pathogenesis before the clinical features of MPX.

Average English

Author Response

The review presented here is all about providing the latest updates regarding the MPX infection.

It appears somewhat superficial and the bibliography is not well structured. In fact, in several cases the references cited in the text do not correspond to the studies listed among the references. In other cases, references are completely missing (for instance, see lines 145-149).

Response: the references have been re-aligned in the bibliography as previously they were mistakenly arranged in the alphabetical order but now correspond to the cited numbering and the missing references have been added.

The section regarding therapeutic options is too concise.

Response: please note that the headings Current Mpox Outbreak (2022) and Future Recommendation Therapeutic options and Potential Vaccines, Therapeutics and Antiviral adjuvant - an Update and discussion includes the updates on therapeutics with futuristic prospective and this portion is approximately 1400 words which is 1/3rd portion of our review. A detail overview has already been covered by us in our previous article on vaccine against mpox in the journal vaccines and therefore here we have kept concise data on therapeutics with more focus on immunogenicity of mpox.

The authors should add more data about the last epidemiological scenario of MPX.

Response: latest data on epidemiology has been added and highlighted in a spate heading of epidemiology.

Finally, it would be more appropriate to describe virology and pathogenesis before the clinical features of MPX. 

Response: the formatting has been done and headings re-aligned and re-numbered.

Reviewer 3 Report

Thank you for sharing your review article on the Monkeypox virus. Here some minor suggested edits and comments that could help to improve the article:

L69-79: Please explain in your manuscript your methodology used in much more detail but at least the search teams as well as inclusion and exclusion criteria applied, and the reviewing process done for articles detected accompanied by the mode of data extraction.

L76: Please justify in your article why you decided to include also review articles. This is rather unusual in the context of preparing a novel review article.

From a more general point of view, I would recommend to outline for potential readers early in manuscript how your article is structured. 

See above.

Author Response

Thank you for sharing your review article on the Monkeypox virus. Here some minor suggested edits and comments that could help to improve the article:

L69-79: Please explain in your manuscript your methodology used in much more detail but at least the search teams as well as inclusion and exclusion criteria applied, and the reviewing process done for articles detected accompanied by the mode of data extraction.

Response: the methodology portion has been modified to incorporate and elaborate upon the required concerns

L76: Please justify in your article why you decided to include also review articles. This is rather unusual in the context of preparing a novel review article.

Response: we have not limited our search criteria and have included wide range of research, reviews, clinical case studies and even letter to the editor to bring about a comprehensive overview. Moreover specifically reviews have also been consulted to get an idea and bring to the readers insights into what’s the latest update on mpox and how it’s been dealt worldwide for its clinical manifestations.

From a more general point of view, I would recommend to outline for potential readers early in manuscript how your article is structured. 

Response: we have added this concern in the methodology section for ease of understating the structuring of our article for potential readers.

Round 2

Reviewer 1 Report

Thanks for your response.

Reviewer 2 Report

The authors have completely answered to all issues or queries.

Fine English

Reviewer 3 Report

Thank you for sharing the revised manuscript. Most of my comments were addressed sufficiently.